# Broken Heartstrings—Post-Traumatic Stress Disorder and Psychological Burden after Acute Mitral Regurgitation Due to Chordae Tendineae Rupture

**DOI:** 10.3390/jcm9124048

**Published:** 2020-12-15

**Authors:** Anna Cranz, Anja Greinacher, Ede Nagy, Hans-Christoph Friederich, Hugo A. Katus, Nicolas Geis, Sven T. Pleger, Christoph Nikendei

**Affiliations:** 1Center for Psychosocial Medicine, Department of General Internal Medicine and Psychosomatics, University Hospital of Heidelberg, University of Heidelberg, 69115 Heidelberg, Germany; Anja.Greinacher@med.uni-heidelberg.de (A.G.); Ede.Nagy@med.uni-heidelberg.de (E.N.); Hans-Christoph.Friederich@med.uni-heidelberg.de (H.-C.F.); Christoph.Nikendei@med.uni-heidelberg.de (C.N.); 2Department of Internal Medicine III, Cardiology, Angiology and Pneumology, University Hospital Heidelberg, Im Neuenheimer Feld 410, 69120 Heidelberg, Germany; Hugo.Katus@med.uni-heidelberg.de (H.A.K.); Nicolas.Geis@med.uni-heidelberg.de (N.G.); Sven.Pleger@med.uni-heidelberg.de (S.T.P.)

**Keywords:** chordae tendineae, acute mitral regurgitation, flail leaflet, heart disease, post-traumatic stress disorder, psychological burden

## Abstract

Chordae tendineae rupture (CTR) is a potentially life-threatening cardiac event often resulting in Acute mitral regurgitation (AMR). We assessed Post-traumatic stress disorder (PTSD), depression, and anxiety symptoms in n=21 CTR patients with AMR (age 82.3 ± 4.2 years; 66.7% men) and compared them to n=23 CTR patients with Chronic mitral regurgitation (CMR) and n=35 Myocardial infraction (MI) patients. Regression analyses revealed that PTSD scores were significantly higher in CTR patients with AMR than in CTR patients with CMR or MI patients. CTR patients with CMR had the lowest levels of PTSD-symptoms. Depression and anxiety scores were elevated across all three groups. Our results suggest that psychosocial factors need to be considered in CTR patients’ care.

## 1. Introduction

Chordae tendineae rupture (CTR) is a potentially life-threatening cardiac event [1]. CTR is characterised by sudden onset, rapid progression of pulmonary edema, hypotension, and left-sided heart failure which may finally lead to severe cardiac shock or pulmonary hypertension and acute right-sided heart failure [2]. CTR predominantly occurs in adult males over 50 [3]. The severity of symptoms depends on the number of ruptured chordae. In mild cases, only single or a few chordae tear with minimal hemodynamic effect, exacting no immediate treatment beyond the long term monitoring of Chronic mitral regurgitation (CMR). However, significant rupture involves the simultaneous abrupt breaking of multiple chords. If the ensuing hemodynamic dysfunctions result in Acute mitral regurgitation (AMR), CTR is classified as a cardiac emergency requiring immediate medical intervention necessitating management through surgical intervention as well as pharmacotherapeutic stabilisation [3]. However, despite the potentially devastating effects on patients’ physical health, the psychological effects of CTR have yet to be described. Moreover, while traumatic conditions within the spectrum of acute coronary syndrome have been at the centre of research efforts, the relationship of post-traumatic stress disorder, depression, anxiety, and CTR has been neglected so far.

The Diagnostic and Statistical Manual of Mental Disorders-Fifth Edition (DSM-5) defines PTSD as a trauma- and stressor-related disorder precipitated by a traumatic event or life-threatening disease. The DSM-5 specifies that a life-threatening disease “is not necessarily considered a traumatic event” per se, but includes “sudden” and “catastrophic” medical events that are experienced as so severe that they qualify as Criterion A traumatic events according to the Diagnostic Criteria for PTSD. Cardinal PTSD symptoms are intrusions, avoidance behaviour, hyperarousal, and negative cognitions and mood. Major cardiac events, and specifically MI- and CTR-induced AMR can enfold uniquely traumatising characteristics [4], like their catastrophic abruptness, the patients’ experience of intense helplessness and fear during the event as well as the high risk of fatality [5]. These traumatic aspects can contribute to the development of acute stress disorder, which can occur less than 1 month after a cardiac event, and/or PTSD or symptoms of PTSD, which can occur at least 1 month after a traumatic event, as has been recently shown for MI [6] and may be the case in CTR induced AMR. Unsurprisingly, previous data have shown that some acute coronary syndromes (MI, unstable angina) patients experience Post-traumatic stress symptoms (PTSS) to the degree of warranting a PTSD diagnosis following a serious cardiac event [7]. To date, studies researching Cardiac-disease-induced post-traumatic stress disorder (CDI-PTSD) suggest that between 12% and 15% of these patients develop PTSD [4,8]. In turn, patients with Cardiovascular disease (CVD) and comorbid diagnosis of PTSD report more risk behaviours, such as smoking, physical inactivity, and obesity [9,10,11]. A meta-analysis [7] found compelling evidence that comorbid PTSD diagnosis in cardiac patients is associated with increased risk of cardiac events and mortality independent of depression. Yet, to our best knowledge, the relationship between CTR and PTSD remains neglected. Somatic diseases are most frequently comorbid with depression and anxiety disorders. Depressive disorders are characterised by persistent depressive mood, anhedonia, and loss of energy. Major depressive episodes have a life-time prevalence of up to 24% [12], placing them among the most common psychiatric disorders. Depression is more prevalent in females and is associated with a significant impairment of quality of life [13] and with high psychological burden including suicidal behaviour [14]. Anxiety disorders are also common [15] and comprise General anxiety disorder, Panic disorder, and phobic disorders. Core anxiety disorder symptomsinclude avoidance behaviour and social withdrawal [16]. General anxiety disorder, Panic disorder and PTSD often co-occur [17] and require specialised treatment as spontaneous recovery is rare [18,19]. Again, specific data evaluating comorbid psychological burden in cardiac patients with CTR is lacking to date. This study, therefore, aimed to assess the prevalence of PTSD, depression, anxiety symptoms in CTR patients with AMR and with CMR, to compare the assessed symptom burden of the CTR sample with the MI sample of similar age and sex, and to investigate variables influencing psychological burden. It was hypothesised that (i) the PTSD, depression, and anxiety symptom burden in CTR patients is increased compared to norm samples; (ii) the psychological burden in CTR patients with AMR is comparable to the symptom burden in MI patients.

## 2. Methods

### 2.1. Study Design and Procedure

For this cross-sectional study, clinical and psychometric data was systematically collected in two cardiological samples (CTR case study patients and MI control study patients) after hospitalisation for a CTR or MI event in the Department of Internal Medicine III, Cardiology, Angiology, and Pneumology, University Hospital Heidelberg, Germany, as detailed in Figure 1. All CTR and MI patients were assessed shortly after a 18-month check-up appointment.

### 2.2. Study Sample

All patients treated for CTR in the period between January 2013 and July 2017 and that were registered for a 18-month follow-up appointment were prospectively included in the study (n=65). Predefined exclusion criteria where: manifest psychotic disorder, bipolar disorder, dementia, drug addiction, and severe comorbid physical conditions, such as cancer or the human immunodeficiency virus. We had to exclude 11 potential participants as they had either died or were no longer able to give informed consent (due to e.g., dementia) when we contacted them for their check-up appointment. All 44 informed consenting patients completed validated psychometric questionnaires assessing their psychological burden (PTSD, depression, and anxiety symptoms) either during a 18-month follow-up appointment or in the comfort of their home in consideration of age- and health-related mobility restrictions.

For the clinical control group, all MI patients over the age of 60 treated on a joint ward of the Departments of General Internal Medicine and Psychosomatics and of Internal Medicine III, Cardiology, Angiology, and Pneumology, University Hospital Heidelberg, Germany, in the period between January 2016 and July 2018 and that were registered for a 18-month follow-up appointment were prospectively included in the study (n=196) following the same exclusion criteria as described above. In order to gain comparable samples with regard to age and sex, we matched eligible MI patients’ age and sex with our CTR sample. We had to exclude 25 potential participants as they had either died or were no longer reachable via telephone when we contacted them for their check-up appointment (see Figure 1). A total of n=35 MI controls provided their informed written consent and completed the same set of questionnaires. However, although n=44 MI patients had agreed their participation when contacted via telephone, only n=35 returning data sets were complete, which equals to 36.5% of MI eligible patients (>60 years old, hospitalised between 2016 and 2018). Hence, we had to exclude nine participants because either their written consent was missing and was not provided until the end of the study period or the data sets had too many missings.

### 2.3. Ethical Considerations

Prior to implementation, the presented study was approved by the University Hospital Heidelberg Ethics Committee (Ethics application no. S-041/2017) and all participants provided their informed written consent. All participants were able to withdraw their consent at any time without fear of any disadvantage to them. The study was conducted in accordance with the Declaration of Helsinki [20].

### 2.4. Measures and Medical Data

Sociodemographic characteristics and medical data: We used a sociodemographic questionnaire to obtain participants’ characteristics as displayed in Table 1. A short psychiatric history was taken, during which we assessed via three self-report items (yes/no) whether patients had ever suffered from an anxiety, depressive, or other psychiatric disorder in their lifetime. The participants’ educational level was divided into low, average, and high levels of education. The patients’ medical data were retrieved from their digital hospital charts and are also shown in Table 1. The left ventricular ejection fraction was determined via transthoracic echocardiographic imaging by taking the average measurements by area length and Simpson’s methods [21]. The angiographic severity of mitral regurgitation graded according to the accepted grading scheme comprising four grades (1,2,3, and 4, from mild to severe) and functional class graded from I to IV by using the New York Heart Association (NYHA) criteria were also retrieved from the medical charts.

Psychometric measures: We assessed primary traumatisation with the German version [22] of the Post-traumatic Diagnostic Scale (PDS). Items are rated on a four point Likert scale (0–3) with total symptom severity scores ranging between 0 and 51. The cut-offs are defined as follows: 0 no rating, 1–10 mild, 11–20 moderate, 21–35 moderate to severe and >36 severe. In line with Kiphuth et al.’s approach in their study on PTSD and transient ischemic attacks [23], we adapted the PDS and defined the CTR or MI event as the A criterion and all DSM-IV criteria *A*–*F* were reviewed accordingly.

We assessed depression using the Patient Health Questionnaire (PHQ)-9 Depression Scale [24], which scores each of the nine DSM-IV criteria from 0 (not at all) to 3 (nearly every day). The severity of depressive symptoms is determined by calculating the sum score ranging between 0 and 27. Scores between 1 and 4 suggest minimal depressive symptoms, scores between 5 and 9 mild depression, scores between 10 and 14 moderate depression, and scores of 15 or above severe depression. The mean sum score for depression was M=3.60(SD=4.08) in a representative German norm sample [25].

To assess anxiety, we used the German version of the Generalised Anxiety Disorder Scale (GAD-7, [26]) which is a 7-item self-report questionnaire that measures anxiety symptoms during the past two weeks. The sum score ranges from 0 to 21: scores between 1 and 4 suggest minimal, scores between 5 and 9 suggest a mild, scores between 10 and 14 suggest a moderate, and scores of 15 or above suggest severe anxiety disorder symptoms. In a study with a representative population sample, the mean sum score of the GAD-7 was M=2.9(SD=3.4) [27].

### 2.5. Data Analysis

All data were coded and analysed using the software package IBM^®^ SPSS^®^ Statistics (version 24) and the statistical package R. Descriptive statistics were used to describe the major study variables and sample demographics, raw data are displayed by showing the Mean (M) and Standard deviation (SD). When normality and homogeneity of variance assumptions were not met, we used the nonparametric Wilcoxon test for the comparison of CTR patients and available norm data. If they were met, we used independent t-tests. Bivariate linear regression analyses were used in three independent models to assess the unadjusted influence of sociodemographic variables (sex, age, civil status, educational level), clinical characteristics (diagnoses of comorbid nicotine abuse and obesity), and assessed patient groups on indicators of psychological burden (1) PTSD, (2) Depression, and (3) Anxiety. Additionally, we performed three different multiple regression analyses in R to determine the relationship between the indicators of psychopathology and assessed patient groups controlled for other confounders. The regression assumptions residual normality, variance homogeneity, and non-multicollinearity (Variance inflation factor (VIF) < 4) were tested for violation with the R-package ‘olsrr’ [28]. If ordinary least square (OLS) regression analyses assumptions were not met, robust regression analyses was used via the R-package ‘robust base’, ‘lmrob’ function [29].

## 3. Results

### 3.1. Sample Description

Overall, n=44 CTR patients and n=35 MI patients completed the questionnaires. This means that 67.7% of all CTR patients hospitalised in Department of Internal Medicine III, Cardiology, Angiology and Pneumology, University Hospital Heidelberg, Germany, between 2013 and 2017 participated in our study. On average, we assessed CTR patients 21 months (range 12–32 months) and MI patients 20 months (range 13–33 months) after hospitalisation. A summary of socio-demographic characteristics are shown in Table 1. Both the CTR and the MI patient group were approximately balanced in sex and of similar age. In line with extant data on prevalence, the CTR patient subgroup with AMR had slightly more males *n* = 14 (66.7%) compared to the subgroup with CMR. In terms of level of education, all groups showed an average level of education, which means that the majority of participants had achieved qualifications in vocational training but only a few had completed higher academic training in the German education system. Across all groups, patients reported a history of previous psychiatric disorders.

### 3.2. Psychological Burden

The means and SD for PTSD, Depression, and General anxiety disorder scores as well as norm data comparisons are presented in Table 2. For PTSD, analysis revealed moderate symptom burden across the CTR group. CTR patients with AMR showed moderate to severe symptom burden, while CTR patients with CMR showed mild PTSD symptom burden. MI patients showed moderate PTSD symptom burden. For depression, analysis revealed moderate symptom burden across all groups. For anxiety symptoms, analysis revealed mild to moderate symptom burden across all groups. We also compared the CTR patients’ psychological burden scores to available norm data. Compared to healthy patients assessed with the PDS [30], our study’s CTR patients’ PTSD scores were significantly higher. We also compared assessed CTR patients’ Depression and anxiety scores to a German norm sample [31] and could show that these were again significantly higher for Depression and for anxiety symptom scores.

### 3.3. Variables Influencing Symptom Scores

The results of the bivariate (unadjusted) linear regression analyses are shown in Table 3, the results of the multiple (adjusted) regression analyses are displayed in Table 4.

Regarding to multiple regression analyses, the inspection of regression assumptions revealed no violation for the PTSD and anxiety disorder models. However, the Depression model did not follow normal distribution (Shapiro−Wilk=0.96,p=0.013;Anderson−Darling=0.87,p=0.025). Accordingly, a robust regression analysis was used for this model. For all regression models, the variance inflation factor was below the value of 3.72. Hence, multicollinearity was not an issue. In the bivariate association, male gender and CTR with AMR group (compared to MI group) were found to be significantly positive and CTR with CMR group (compared to the MI group) significantly negative predictors of PTSD symptoms. Multiple regression analysis indicated that the added variables were able to explain 49% of the variance in PTSD ratings. Here, by controlling for other variables, only the two CTR group variables proved to be significant. In the depression prediction models, only the female sex was significant in bivariate regression analyses. In the multiple regression model, 16% of the variance could be explained by the predictors. Even in the adjusted model, only female gender significantly predicted depression.

## 4. Discussion

To the best of our knowledge, this study is the first to document psychological burden, namely the prevalence of PTSD, depression, and anxiety symptoms, in cardiac patients with CTR using established questionnaires. Furthermore, this study is the first to compare CTR patients with AMR’s psychological burden to the mental distress reported by patients with CMR and patients with MI of similar age and sex as well as to examine variables potentially influencing psychological burden via regression analysis. Our data analysis revealed important findings: first, as hypothesised, all CTR patients’ PTSD, depression, and anxiety symptom burden was significantly elevated compared to corresponding norm sample data. Furthermore, our data shows that assessed CTR and MI patients were significantly burdened by PTSD symptoms reporting moderate to severe symptom burden scores. As anticipated, CTR patients who had suffered AMR reported significantly more PTSD symptoms compared to CTR patients with CMR. Our data revealed CTR patients who had suffered AMR showed the highest PTSD scores across all assessed groups with the reported symptom burden nearly reaching the PDS cut-off score for severe PTSD symptoms (>30). While CTR patients with CMR showed the lowest PTSD symptoms.

### 4.1. Post-Traumatic Stress Disorder

With regard to PTSD symptom load, our data suggests that CTR patients with AMR’s psychological burden not only exceeds the mental distress reported by norm samples [32], but also the mental burden of the MI patients in our study. CTR patients with CMR reported significantly less PTSD symptom load compared to CTR patients with AMR and MI patients. Overall, our results support the outcomes of previous studies assessing comorbid psychological burden in cardiac patients [33,34] and provide first valuable insights into the nature of psychological burden in the wake of CTR, especially with regard to PTSD symptom burden.

Male gender and a more severe degree of mitral regurgitation were significant predictors of PTSD symptoms. This could be explained by the fact that CTR induced AMR is more prevalent in men and that gender-related differences still exist in emotion regulation. For example, alexithymia, which refers to a deficit in processing emotions including difficulties in identifying feelings, describing feelings, and processing emotions in externally oriented thinking, has been shown to play a role in PTSD development in men [35]. Furthermore, our data suggests that comparatively younger (bear in mind we assessed an octogenarian sample), more obese; and CTR patients with AMR were more likely to report a high PTSD symptom load. However, this marginally significant effect must be interpreted with caution and a larger sample is needed to verify its influence. Many studies have found evidence suggesting that PTSD is associated with an increased risk for obesity due to perceived high chronic stress and resulting stress response system dysregulations [36,37,38]. In turn, poor stress-related health behaviours are well established contributors to known cardiometabolic risk factors [39]. In their 2019 review, Aaseth et al. [36] underline the importance of addressing somatic correlates of PTSD and suggest a combined therapeutic approach of dietary re-regulation as well as pharmaco- and cognitive psychotherapeutic therapy with graduated desensitisation toward triggering factors. However, although ample valid approaches exist, this highly somatically and psychologically burdened group of elderly patients is likely to require more tailored psychotherapeutic interventions compared to affected adults under 60 [40]. Notably, Ditlevsen and Elklit [41] found that men in their late 60s showed the lowest potential risk of PTSD in their comprehensive review of Danish and Nordic studies of PTSD or trauma. However, their review also revealed that the risk of PTSD increases between the late 60s and the late 70s for men and from the mid to late 70s for women. It has been argued that octogenarians face special challenges and may be in a new stage in psychosocial development in which they must deal with the acceptance of earlier experiences in life and the fact that death is more imminent than earlier [42]. Furthermore, childhood trauma is associated with more PTSD symptoms and greater impairment in older age compared to trauma in adult age [43]. Although this pilot study did not systematically assess previous traumatic events, we must assume that participants were either survivors of war-related childhood trauma or at least effected by trans-generational war-related effects [44]. We did, however, assess the study’s patients’ history of previous psychiatric disorders via three self-report items. These descriptive data point to the fact that this study’s patient samples were pre-burdened by PTSD, depressive, and anxiety symptoms. However, only few assessed patients reported a history of PTSD symptoms prior to the cardiac event. In non-medical settings, previous trauma exposure and pre-existing PTSD have been shown to be key predictors of PTSD after further trauma [45]. However, some authors argue that medical event induced PTSD may differ from more traditional PTSD concepts because the trauma originates inside the body (e.g., a blocked coronary artery, CTR). Hence, studies assessing these patient samples have found that patients were often preoccupied with fears related to the recurrence of events originating inside the body as opposed to memories of past trauma [46]. Moreover, prior to exposure to an acute, life-threatening medical event, individuals suffering from a cardiac disease may be more prone to developing PTSD symptoms due to stress sensitisation from related experiences [47]. Hence, patients experiences of their cardiac history should also be assessed in future research. In contrast, trauma from non-medical events may also predispose patients to PTSD symptoms after medical events due to a more generalised disruption of the psychological and neurobiological systems responsible for recovery from trauma. This further underlines the need to systematically assess previous psychiatric history.

### 4.2. Associations between PTSD and Assessed Somatic Functional Parameters

Although our data is cross-sectional, our results show AMR patients report experiencing severe emotional stress reflected in the significant association of high PTSD scores and severe cardiac symptoms during AMR after CTR. This relationship between psychological burden and symptomatic response has been observed in patients with cardiac [48] and other conditions, such as pulmonary disease [49], asthma [50], and cancer [51]. This study is the first to document this relationship in CTR patients.

The fact that the severity of somatic symptoms in the assessed patients predicted PTSD-pathology is seemingly in contrast to the findings of Bayer-Topilsky et al. [52]. Although their comprehensive study was able to demonstrate that PTSD is also prevalent in patients with CMR, they argued that PTSD was not determined by objective mitral-regurgitation severity or consequences but was rather linked to experienced feelings of anxiety and depression than to somatic cardiac symptoms, such as dyspnea. While this may hold true for in the assessment psychological symptom burden for patients with CMR, our data suggests that this explanation may not be applicable for CTR patients with AMR as the patients’ somatic experiences of the two diseases are very different with regard to immediate perceivable impact. While CMR is associated with progressive, long-term loss of quality of life, CTR with AMR is a highly dramatic, life-threatening event which is reflected in the high PTSD symptom scores reported in this study. Furthermore, the difference between PTSD as a pre-existing comorbidity with negative prognostic impact and PTSD as a cardiac-induced disease resulting from an immediate and potentially fatal cardiac event needs to be considered. In their comprehensive review, Vilchinsky et al. [4] argue that in the case on cardiac-disease-induced PTSD, namely when criterion A according to DSM-V is the cardiac event itself as in this study, the severity of cardiac disease is a risk factor. Our results can be seen to corroborate Vilchinsky et al. suggestion that CDI-PTSD may prove to be a conceptually and empirically diagnostic entity worthy of further consideration, especially with regard to psychotherapeutic treatment needs.

### 4.3. Depression

With regard to depression, our data revealed that all CTR patients averagely reported mild to moderate depression scores entailing severe emotional distress up to, in part, suicidal ideation [53]. The mean scores for depression reached mild to moderate cutoff scores across all groups which is comparable to other patient groups with somatic diseases and comorbid depression [54,55]. All CTR patients showed elevated depression scores compared to norm samples [31]. Bivariate regression analysis revealed that female sex was significantly associated with higher depression scores. This was also the only significant predictor in multiple regression analyses. This is in line with previous findings as it is known that depression is more commonly reported by females than males [56]. Depression is three to five times more prevalent in heart failure patients than in the general population and again prevalence rates are also higher in females here [57].

The American Heart Association officially recognises depression as risk factor for poor prognosis among patients following acute coronary syndromes [58] and underlines that the diagnosis and treatment of comorbid depression is key element of best-practice clinical care. In addition, European guidelines identify depression, anxiety, and psychosocial stressors, such as work-related stress or poor social support, as risk factors for incident CVD and adverse outcomes in patients with existing CVD [59]. However, despite recent efforts, fewer than 25% of cardiac patients with major depression are diagnosed as depressed, of which only half receive depression treatment [60]. This is reflected in our data, with only 4 patients reporting to receive regular pharmacological treatment to address their depressive symptoms and none getting regular psychotherapeutic treatment. Research investigating psychotherapeutic treatment referrals of elderly patients in Germany has shown that, although psychotherapy is covered by German health insurance, older people with depressive disorders access psychotherapy far less frequently than younger patients and are more likely to have anachronistic, stigma-biased conceptions about it [61]. This is alarming, as there is effective treatment for depressive disorders with current guidelines specifying either psychotherapy or antidepressant treatment of moderate depression and a combination of both for severe depression.

### 4.4. Anxiety

With regard to anxiety, our data suggest that on average, CTR patients reported mild to moderate general anxiety symptoms including feelings of threat, restlessness, irritability, insomnia, tension, and physical symptoms, such as palpitations. This is comparable to other patient groups with somatic diseases displaying comorbid anxiety [62,63]. All CTR patients showed higher anxiety scores compared to norm samples [27]. Overall, the CTR patients in our study had slightly higher anxiety scores than assessed MI patients. However, our study is unable to provide evidence of variables influencing anxiety scores.

Research regarding cardiovascular prognosis assessing the influence of general anxiety disorder has yielded inconsistent results. Some studies have demonstrated that anxiety is as an etiological risk factor of adverse cardiovascular events [64], such as myocardial infarction [65] and ischemic stroke [66]. Other large scale studies have revealed results suggesting that patients with elevated general anxiety showed a better prognosis following a cardiac event [67,68] and argue that increased anxiety may result in more alertness and health promoting behaviour [69]. In turn, this may be reflected in our results pointing towards increased health behaviour in patients with higher anxiety symptoms showing less nicotine abuse in our study. Future research should address the predictors and implications of prevalent anxiety disorders more comprehensively in CTR patients in a longitudinal design with a larger sample size.

## 5. Conclusions

Although our study has the great advantage of being the first study to investigate the psychological burden in CTR patients, it is limited by the small sample size, low statistical power, and cross-sectional design which, in turn, also restricted the number of influencing variables which could be included in regression analysis. Regarding our results, it must be noted that low statistical power negatively affects the likelihood that a nominally statistically significant finding actually reflects a true effect. A larger sample would have enabled us to adjust for confounders and improve stratification. Regarding anxiety symptoms, our study may have lacked enough power to detect possible differences. Due to the cross sectional design, causal conclusions cannot be drawn and longitudinal analyses are required. In addition, pre-cardiac event psychiatric history was not systematically measured. Furthermore, limiting generalisability, it must be assumed that the specific pathology profile of this elderly sample assessed monocentrically may differ from the pathology profile of younger CTR patients assessed across several centres, especially with regard to number of diagnosis and resulting variable interference as well as war-generation related confounding factors. Further potential influencing factors need to be addressed in a more comprehensive assessment of sociodemographic, medical, and psychological data, such as attachment style and personality traits in a longitudinal design in a younger larger sample. Also, a systematic recruitment bias cannot be ruled out which could have led to lower anxiety disorder prevalence due to avoidance behaviour, or in contrast, higher anxiety disorder prevalence due to reassurance seeking behaviour.

Nevertheless, our study has shed light on the so far largely neglected psychological burden of cardiac patients with CTR and presented results point to the fact that there is, indeed, a need for considering psychosocial factors in the care of CTR patients. Especially, with regard to the high prevalence of PTSD symptoms in CTR patients with AMR, the presented data suggests the necessity for meaningful interventions to treat and prevent post-traumatic stress disorder after CTR.

## Figures and Tables

**Figure 1 jcm-09-04048-f001:**
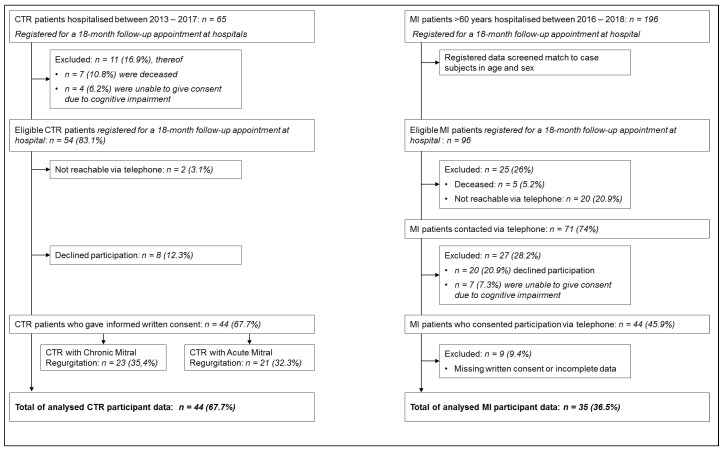
Chordae tendineae rupture (CTR) and Myocardial infraction (MI) patient sample recruitment; percentages (%) from the initial number of cases and are shown in parentheses.

**Table 1 jcm-09-04048-t001:** CTR and MI patients’ sociodemographic characteristics. The CTR patients’ data are displayed as a whole and in two subgroups: “CTR with AMR” and “CTR with CMR.”

	All CTR Patients *n* = 44		CTR with AMR *n* = 21		CTR with CMR *n* = 23		Myocardial Infraction *n* = 35	
	*n* (%)	M (SD)	*n* (%)	M (SD)	*n* (%)	M (SD)	*n* (%)	M (SD)
Sex								
Male	25 (56.8)		14 (66.7)		11 (47.8)		21 (60)	
Female	19 (43.2)		7 (33.3)		12 (52.2)		14 (40)	
Age (years)		81.9 (5.1)		82.3 (4.2)		81.6 (5.9)		78.9 (5.1)
Civil status								
Married	7 (15.9)		4 (19)		3 (13.0)		10 (28.6)	
In a relationship	27 (61.4)		11 (52.4)		16 (69.6)		14 (40.0)	
Widowed	10 (22.7)		6 (28.6)		4 (17.4)		11 (31.4)	
Educational level (scale 1 to 5)		2.7 (1.1)		2.7 (0.9)		2.8 (1.2)		3 (1.2)
Clinical parameters								
LVEF %		30.0 (11.9)		27.5 (9.2)		34.4 (15.1)		
LVEDD		59.9 (10.8)		61.9 (8.4)		57.9 (12.7)		
6 Min Walking Distance [s]	37 (84.1)	346.8 (97.3)	21 (100)	350.2 (82.2)	16 (69.6)	342.4 (116.9)	8 (22.9)	438.5 (131)
Degree of mitral regurgitation		3.1 (0.5)		3.3 (0.7)		2.1 (0.9)	2 (5.7)	2 (1.4)
NYHA (scale 1 to 5)		2.9 (0.5)		2.8 (0.6)		2.9 (0.5)	2 (5.7)	1.5 (0.5)
Additional diagnoses								
Nicotine abuse	27 (61.4)		14 (66.7)		13 (56.5)		26 (74.3)	
Diabetes	24 (54.5)		10 (47.6)		14 (60.9)		24 (68.6)	
Obesity	25 (56.8)		11 (52.4)		14 (60.9)		20 (57.1)	
History of								
Anxiety disorder	15 (34.1)		8 (38.1)		7 (30.4)		13 (37.1)	
Depressive disorder	17 (38.6)		8 (38.1)		9 (39.1)		16 (45.7)	
PTSD	7 (15.9)		3 (14.3)		4 (17.4)		5 (14.3)	

Notes: CTR = Chordae tendineae rupture; AMR = Acute mitral regurgitation; CMR = Chronic mitral regurgitation; M = mean; SD = standard deviation; LVEF = Left ventricular ejection fraction; LVEDD = Left ventricular end-diastolic diameter; NYHA = New York Heart Association Classifications; When *n* (%) and M(SD) are shown, data was not available from all participants for that parameter and n refers to how many data were available in that subgroup.

**Table 2 jcm-09-04048-t002:** CTR and MI patients’ psychometric variables for PTSD, Depression, and Anxiety disorder scores as well as norm data comparisons. The CTR patients’ data are displayed as a whole and in two subgroups: “CTR with AMR” and “CTR with CMR”.

	All CTR Patients		Norm Data Comparisons							CTR Patients with AMR		CTR Patients with CMR		MI Patients	
	*n* = 44		*n* = 1201 ${}^{1}$			*n* = 50,362 ${}^{2}$				*n* = 21		*n* = 23		*n* = 35	
			Wilcoxon test			T-Test									
	M	SD	M	SD	*p*	M	SD	T(df)	*p*	M	SD	M	SD	M	SD
Psychometric variables															
PTSD score	21.40	12.5	12.54	10.54	<0.001					29.81	6.91	13.74	10.78	22.94	4.99
Depression score	7.84	4.05	5.9	4.2	<0.001					6.76	3.83	8.82	4.07	7.60	4.70
Anxiety score	9.91	4.24				2.95	3.41	t(43)	<0.001	9.90	4.08	9.91	4.48	8.60	4.91

Notes: CTR = Chordae tendineae rupture; AMR = Acute mitral regurgitation; CMR = Chronic mitral regurgitation; M = mean; SD = standard deviation; t = t-value; *p* = *p*-value as indicator of statistical significance; PTSD = Post-traumatic stress disorder; ${}^{1}$ healthy patients assessed with the PDS [30]; ${}^{2}$ German norm sample [31].

**Table 3 jcm-09-04048-t003:** Results of bivariate linear regression analyses of predictors on indicators of psychological burden.

	Bivariate Model for PTSD					Bivariate Model for Depression					Bivariate Model for Anxiety Disorder				
	B	Se (B)	β	*p*	Wald	B	Se (B)	β	*p*	F/Wald	B	Se (B)	β	*p*	F/Wald
Age	−0.29	0.24	-	0.230	1.47	0.09	0.09	0.11	0.353	0.42	0.08	0.09	0.09	0.400	0.71
Sex	−4.62	2.31	-	0.048	4.02 *	3.28	0.98	-	0.001	11.28 ***	0.80	1.27	-	0.534	0.39
Civil status					1.81					0.74					0.04
in relationship	−2.51	2.81	-	0.375		−0.78	1.61	-	0.631		−0.33	1.33	−0.37	0.803	
widowed	0.92	3.09	-	0.766		−1.32	1.61	-	0.416		−0.10	1.33	−0.01	0.947	
Educational level					0.38					1.92					0.79
middle	−0.42	5.35	-	0.937		0.44	1.99	0.04	0.826		1.12	2.13	0.11	0.601	
high	1.22	5.61	-	0.829		−1.92	2.19	−0.18	0.382		2.50	2.35	0.22	0.290	
Nicotine abuse	4.35	2.56	-	0.094	2.87 †	0.64	1.04	0.07	0.539	0.38	−1.23	1.09	−0.13	0.264	1.27
Obesity	−3.72	2.27	-	0.105	2.69	−0.44	1.02	-	0.666	0.19	0.01	1.04	0.01	0.993	0.00
Patient groups					101.92 ***					2.92					0.80
CTR with AMR	7.39	1.99	-	<0.001		−0.74	1.18	-	0.534		1.30	1.23	0.12	0.289	
CTR with CMR	−14.55	1.56	-	<0.001		1.36	1.22	-	0.265		1.31	1.23	0.13	0.289	

Notes: PTSD = post-traumatic stress disorder; AMR = acute mitral regurgitation; CMR = chronic mitral regurgitation; B = unstandardised regression coefficient; se = standard error, β = standardised coefficient (only for OLS-Regression analyses); *p* = *p*-value as indicator of statistical significance; F/WALD = Bivariate model fit statistics with significant levels: *** = *p* < 0.001, * = *p* < 0.05, † = *p* < 0.1; reference category for civil status: married; reference category for educational status: low; reference category for the CTR with AMR patient group and the CTR with CMR patient group: the MI patient group.

**Table 4 jcm-09-04048-t004:** Results of multiple linear regression analyses of predictors on indicators of psychological burden.

	Model for PTSD				Model for Depression				Model for Anxiety disorder			
	**B**	**Se (B)**	β	***p***	**B**	**Se (B)**	β	***p***	**B**	**Se (B)**	β	***p***
Age	−0.33	0.17	−0. 18	0.059	0.05	0.08	-	0.553	0.08	0.11	0.10	0.445
Sex (male = 0)	−2.81	1.85	−0.14	0.133	2.85	1.12	-	0.020	0.86	1.17	0.09	0.468
Civil status							-					
in relationship	−1.34	2.22	−0.07	0.549	−0.62	1.39	-	0.657	−0.80	1.41	−0.09	0.572
widowed	−1.13	2.52	−0.05	0.657	−0.35	1.46	-	0.808	−0.13	1.60	−0.01	0.938
Educational level							-					
middle	1.12	3.62	−0.05	0.758	−0.18	2.22	-	0.934	0.57	2.30	0.05	0.807
high	0.97	3.88	0.04	0.803	−1.30	2.20	-	0.555	2.77	2.47	0.24	0.267
Nicotine abuse	2.50	1.83	0.12	0.178	0.95	1.06	-	0.377	−1.08	1.17	−0.11	0.360
Obesity	−3.32	1.78	−0.17	0.065	0.09	1.04	-	0.933	0.09	1.13	0.01	0.987
Patient groups												
CTR with AMR	7.97	2.15	0.37	<0.001	−0.81	1.28	-	0.530	1.33	1.37	0.13	0.334
CTR with CMR	−7.14	2.12	−0.34	0.001	0.86	1.29	-	0.507	1.21	1.35	0.12	0.375
Model Summary												
Model F (df1, df2)	6.51 (10, 68); *p* <0.001									0.58 (10, 68); *p* = 0.822			
Wald test—$X^{2}$ (df1, df2)					17.09 (68, 78); *p* = 0.031							
$R^{2}$ (adjusted $R^{2}$)	0.49 (0.41)				0.17 (0.05)				0.08 (−0.06)			

Notes: PTSD = post-traumatic stress disorder; AMR = acute mitral regurgitation; CMR = chronic mitral regurgitation; B = unstandardized regression coefficient; Se = standard error, β = standardized coefficient (only for OLS-Regression analyses); *p* = *p*-value as indicator of statistical significance; $R^{2}$ = determination coefficient; reference category for civil status: married; reference category for educational status: low; reference category for the CTR with AMR patient group and the CTR with CMR patient group: the myocardial infraction patient group.

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
