# Peer review of "Broken Heartstrings—Post-Traumatic Stress Disorder and Psychological Burden after Acute Mitral Regurgitation Due to Chordae Tendineae Rupture"

_jcm, 2020, doi:10.3390/jcm9124048_

Round 1
Reviewer 1 Report
This is an interesting and well-done study. I have some major concerns about the psychiatric portions which are neither adequately clarified nor discussed.
The question of whether a life-threatening illness meets PTSD’s Criterion A is complicated and nuanced. According to DSM-5
“A life- threatening illness or debilitating condition is not necessarily considered a traumatic event. Medical incidents that qualify as traumatic events involve sudden, catastrophic events (e.g., waking during surgery, anaphylactic shock).”
In other words, the medical event must occur under traumatic circumstances to qualify. Therefore, the authors need to include some description of what the subjective emotional experience of CTR with AMR is like so that readers can be confident that Criterion A is met. Furthermore, there is abundant evidence that subjective reports by (some, but not all) MI patients indicate that they may also experience their Mis as catastrophic events that might meet criterion A. This would explain why PTSD severity in MI patients (Table 3) is also in the moderate-to-severe range (although not as high as the STR/AMR patients.) If neither CTR/AMR nor MI meets Criterion A during the acute event, they cannot receive a PTSD diagnosis and subsequent PTSD scores simply indicate the severity and qualitative nature of their distress but cannot retroactively enable one to diagnose PTSD. This is extremely important and the authors need to do a much better job addressing this point.
Although the authors mention that this is a cross-sectional study, I believe that they need to devote more attention to this fact because one really cannot infer the direction of causality from their findings.
Related to the last point, lack of a trauma history or psychiatric history really muddies the inferential waters. How many CTR/AMR (or MI or CTR/CMR) had prior trauma exposure or even PTSD before the critical cardiac problem. This is crucial information if we are to disentangle other variables that might contribute to their findings. This is mentioned in passing (Lines 235-237) but, again, deserves a more thoughtful discussion.
Two minor points:
Line 27 – DSM-5, not DSM-V
Line 29 – add negative thoughts & cognitions to the list of PTSD symptom clusters (e.g. “intrusions, avoidance behavior and arousal”)
Author Response
Dear Reviewer,
Please see the attachment for the full coverletter and replies to all the points raised - Thank you for your valuable imput!
Kind regards
Anna Cranz

Reviewer 2 Report
Thank you for letting me review the paper Broken Heartstrings - Post-Traumatic Stress Disorder and Psychological Burden after Acute Mitral Regurgitation due to Chordae Tendineae Rupture – a Pilot Study. The topic is very interesting as the emotional aspects of heart disease is still not well understood.
The manuscript is generally language-wise and in the introduction well written but still I have some major concerns:
- This is not a “pilot study”. It is just an underpowered study. Nowhere is a larger study mentioned that this study is supposed to pilot. A pilot study should focus on feasibility outcomes and perhaps say something preliminary about effect sizes just to guide the sample sizes of the main study. This study draws conclusions from inference statistics which is not in line with pilot study methodology. As this study includes patients for several years, a sufficiently powered study may not be feasible with the present design.
- In the Background it is stated that “GAD, PD and PTSD often co-occur and require specialised treatment as spontaneous recovery is rare” with a reference “19” to a study about “Alveolar Echinococcosis”. As it is a strong claim one would have expected a strong reference. Even though I haven’t read it I doubt this source really can confirm the statement.
- There are 4 hypotheses of which “iii” includes 4 sub-hypotheses. If you plan to test all these hypotheses, you must correct for multiple comparisons. Alternatively you just point out your main interest. Based on the introduction there is a more specific interest in CTR than what this list of hypotheses reflect. Further, how is hypothesis ii and iv different? Rewrite this section with clarity.
- The Flow-chart raises more questions than it explains. The steps through which the patients are recruited seems not to be parallel. E.g. for CTR the diseased are not eligible but MI-patients are first matched and then the diseased are excluded. Did they actually die after being included in the study? In neither of the sides excluded patients based on exclusion criteria are specified, why? E.g. the 25 in the MI-group that are excluded after the group have been matched to the CTR-group. The methodology in this part is not convincing or clearly described.
- In the Flow-chart it is stated that 23 has CTR with AMR. In the rest of the article these are counted as 21. I think that the figures have been reversed somehow.
- A very important information is when in a timely relation to the respective event were the psychological measurements made? Shortly after? Was it the same time after the event for all groups? This is essential information.
- Statistical methods: Md is not used for Mean.
- An assumption can be met (not given). I cannot see that all the measures of correlation was used.
- There are much text in the Results that should be moved to the Methods. All the reasoning about missing data and statistical analyses should be in the Methods. Also, avoid statements about surprise or other interpretations in the Result section. Keep them for the Discussion.
- Table 1 (and 2, 3 & 4): You do not need footnotes to explain abbreviations. Just explain all the abbreviation below the table in a note without numbers.
- Table 1: Some variables has both “n (%)” and “M (SD)”. This is difficult to understand. Does the n stands for those above a threshold or what?
- The comparisons with norms are sometimes done with a non-parametric test and at another time with a t-test without any explanation (that belongs in the Methods section by the way). At one time both the compared Means are mentioned and another time only the norm is mentioned (as the clinical group’s value is in the Table). Consistency is warranted. These comparisons could also have been added to Table 2.
- Page 6 row 169: Why is one section that should definitely be part of the Results numbered “4” as it was on the same level as Background, Methods, Results and Discussion?
- Table 3. With univariate you obviously mean bivariate.
- Table 3. This is confusing. It seems like the numbering in the table head and column 1 is not the same. Otherwise how can the same variable be correlated with itself?
- Is Table 3 really necessary? It is a very complicated way of showing how the variables are related and most of the relationships are not interesting for the reader.
- Table 4 and related text.
- Why spend so much effort on the total model statistics as they have nothing to do with your hypotheses?
- Unstandardized betas is almost always most interesting and most interpretable with these kinds of research questions. Her you use the standardized beta for two models and unstandardized in one. Please be consistent and give preferably unstandardized betas.
- Adding bivariate (crude) linear regression analyses accompanying all the three models would be much more interesting than Table 3 and would in important aspects replace it.
- You don’t state the reference for sex, civil status or educational level. Wasn’t there 3 levels of education (or even 5)? Then we should see two estimates. Civil status has also three categories according to Table 1. Then it should be 2 estimates, otherwise one could expect that you treat them as linear and that would be wrong.
The study's methodology and resluts have to be revised before the Discussion can be reviewed.
Author Response

(The authors gave the same response as above.)

Round 2
Reviewer 1 Report
This is an excellent revision. You have addressed all of my concerns.
Author Response
Dear Reviewer 1,
Please see the attachment for the full cover letter response.
Kind regards
Anna Cranz

Reviewer 2 Report
The authors responded well to my comments and I find the methods and presentation of results much improved. It is still a small study with a limited statistical power which is a major limitation.
I have some detailed comments:
- I think that the finding that CTR-CMR had lower levels of PTSD-symptoms should be reflected also in the abstract, even though it was not part of the hypothesis, as it was one of the clearest results.
- In Table 3 the word "Model" is used for bivariate/unadjusted analyses. A "Model" typically refers to an adjusted analysis. Consider using the word "Crude" or "Unadjusted" instead.
- I'm awere that I brought up the word "underpowered". However something can only be underpowered if there are no results, as it refers to the Type 2 error that you can't make if you have a result. This study is only powered to find really big differences, which you did. Discuss instead the general problems with a small sample size such as possibilities to stratify and control for confounders (which you already do, but consider the wording). Regarding Anxiety, power might have been a problem worth mentioning though, but evidently not regarding PTSD-symptoms.
- Generally I think the Discussion is a bit long and that 93 references are way to many. If cuts could be made the manuscript would be improved.
Author Response
Dear Reviewer II,
Please see the attachment for the full cover letter response.
Kind regards
Anna Cranz
